# Adjunctive Plasma Rich in Growth Factors in the Treatment of Osteomyelitis and Large Odontogenic Cysts Prior to Successful Implant Rehabilitation: Case Report

**DOI:** 10.3390/dj11080184

**Published:** 2023-07-31

**Authors:** Marc DuVal, Mohammad Hamdan Alkhraisat

**Affiliations:** 1Department of Dentistry, McGill University, Montreal, QC H3A 0C7, Canada; 2Montreal Jewish General Hospital, Department of Dentistry and Oral and Oral and Maxillofacial Surgery, Montreal, QC H3T 1E2, Canada; 3Santa Cabrini Hospital, Department of Surgery, Division of Plastics, Montreal, QC H1T 1P7, Canada; 4Regenerative Medicine Department, BTI Biotechnology Institute, 01007 Vitoria, Spain; mohammad.hamdan@bti-implant.es

**Keywords:** tissue engineering, bone regeneration, platelet-rich plasma, osteomyelitis, odontogenic cysts, osseous pathology, nerve injury

## Abstract

Osteomyelitis of the jaws presents a clinical challenge to conventional treatment, often requiring multiple surgical interventions resulting in significant deformity and presenting significant problems to satisfactory rehabilitation. While benign odontogenic cysts, such as the radicular cyst, are generally predictably treated, they can cause significant localized bone destruction and thus can present significant problems in satisfactory rehabilitation. In this case report, patients were treated combining conventionally appropriate surgical debridement and oral antibiotics with adjunctive plasma rich in growth factors (PRGF). Patients showed a complete soft tissue and osseous regeneration to their pre-pathologic state, with successful implant rehabilitation. PRGF appears to be highly successful in minimizing or eliminating osseous deformities normally associated with conventional treatment of osteomyelitis of the jaw and treatment of large odontogenic cysts. Future trials must be performed to confirm these results in comparison to controls using conventional treatment alone.

## 1. Introduction

Osteomyelitis is a condition that may present as osseous necrosis, sequestrae, osteolysis, or condensing osteitis. The etiology of osteomyelitis is most commonly due to opportunistic bacterial infection in a susceptible or compromised host or where local factors impair tissue vascularity or host response. Common systemic factors include diabetes, smoking, poor general health, or taking systemic steroids [1,2,3,4]. Osteomyelitis initiation and progression results in osseous tissue injury and death, resulting in vascular compromise and impaired host healing response [1,2,3,4]. Bacterial proliferation within poorly vascular and necrotic bone adds to disease burden and the formation of fistulae can result. Maxillofacial osteomyelitis presents significant clinical challenges. Treatment options range from conservative (palliative, antibiotics, or limited curettage and debridement to vital bone and use of antibiotic beads) to more aggressive (saucerization, decortication, or resection) [1,2,3,4]. Outcomes are unpredictable and often depend on clinical presentation, systemic factors, and presence of co-morbidities such as diabetes, smoking status, periodontal disease, and medications such as steroids or immunomodulators [1]. Treatment often requires long-term, broad-spectrum antibiotics. The use of peripheral indwelling central catheters is not uncommon, thus requiring an additional procedure, hospital human resources, and cost. More recently, bisphosphonate therapy has been used to treat diffuse sclerosing osteomyelitis [4,5]. Unfortunately, bisphosphonates are well known to predispose patients to medication-related osteonecrosis of the jaw (MRONJ), particularly in the context of surgical, endodontic, or periodontal manipulation [6,7,8,9].

Medullary bone destruction results in localized vascular compromise that limits both effective host response to the infectious process as well as limiting repair of injured osseous tissue [10,11,12,13,14]. Moreover, the effectiveness of antibiotics is similarly limited due to poor vascularity of the affected bone. Inflammatory mediators, cytokines, and interleukins contribute to further localized tissue injury, which only further complicates host response and contributes to disease progression [15]. Long-term antibiotics or placement of antibiotic beads can contribute to resolution in some cases, but unfortunately, this remains unpredictable [12,13,16]. Antibiotics certainly do not address the underlying host response nor the underlying localized, compromised vascularity. Moreover, pathologic fracture of the mandible or extra oral fistula formation are serious complications that can result in severe cases [17,18]. In some cases, vascularized flap reconstruction harvested from the tibia is required to manage pathologic fracture or to reconstruct post segmental resection [19]. Conventional goals of treatment are maintenance of osseous continuity, re-establishment of mucosa and skin soft tissue coverage, maintenance or return of function, and resolution of pain and symptoms [2]. Even when successfully treated, satisfactory rehabilitation with implants is often impossible. In many cases, satisfactory conventional denture fabrication is impossible.

The adjunctive use of plasma rich in growth factors (PRGF) could provide a transient fibrin scaffold and biomolecular cues to the resident cells to trigger the healing [20]. PRGF has reported to have an anti-inflammatory effect that would reduce the intensity of the inflammation and accelerate the transition to the proliferative phase where the resident cells remodel the fibrin scaffold to proper provisional matrix that would orchestrate the evolution toward tissue healing [21]. Indeed, platelet-rich plasma has been used as rescue therapy to treat a patient with chronic osteomyelitis (4 years) [22], though no reports in the literature exist on the use of platelet-rich plasma in the treatment of osteomyelitis of the jaw. Platelet-rich plasma has also been used in treatment of non-specific odontogenic cysts [23]. For example, PRGF has been used in the surgical treatment of radicular cysts to enhance bone healing [24]. Although radicular cysts are predictably treated with surgical excision [25,26,27], large size radicular cyst may result in thin, poorly vascularized bone at its periphery [28,29]. This thin bone will routinely be resorbed, and thus, conventional surgical treatment of large radicular cysts will often result in large osseous defects. These defects can make satisfactory rehabilitation problematic or at times impossible.

No reports could be found where PRP or PRGF has been used to treat osteomyelitis of the jaw, nor could any studies be found in which odontogenic cysts were treated with PRP or PRGF resulting in maintenance of pre-pathologic osseous bone volume and subsequent successful implant rehabilitation. In this case report, the main objective has been the description of the clinical outcomes of the use of PRGF in a clinical protocol to treat osteomyelitis and large odontogenic cysts of the mandible and subsequent oral rehabilitation with implant-supported prostheses. The findings of this study would help the clinicians to establish a clinical protocol on how to use plasma rich in growth factors in these challenging clinical cases. In all cases, PRGF was offered to the patient for use as an adjunct to the current standard of care, with involvement of other medical services where appropriate. PRGF and platelet-rich plasma is approved by Health Canada as a “drug” for use in promotion of wound healing, and thus, no institutional review board (IRB) approval was necessary. The results of the use of PRGF as an adjunct for clinically challenging cases are presented in the hopes that it may bring attention to the potential of PRGF to improve clinical outcomes.

The purpose of using adjunctive PRGF in these cases was to augment host healing response and to attempt to maximize osseous regeneration. The purpose of the case series is to demonstrate the apparent benefits of use of adjunctive PRGF in treatment of non-neoplastic pathology, specifically odontogenic cysts and osteomyelitis.

## 2. Materials and Methods

All cases were treated by a single Oral and Maxillofacial Surgeon (MDV) in an outpatient setting under local anesthetic. The McGill University Institutional Review Board was consulted and determined that no ethics review was necessary for publication as PRGF is approved in Canada for wound healing and was used as an adjunct to conventional treatment. The case series inclusion criteria were all cases treated by a single surgeon where PRGF was used in treatment of osteomyelitis and odontogenic cysts over a 18 month time frame.

Patients with odontogenic cysts and osteomyelitis of the mandible were surgically treated by enucleation of the lesion cyst or conservative debridement and curettage, with removal of sequestrate followed by the application of the PRGF. In a single case aseptic osteomyelitis limited to the cortical bone of the inferior border of the mandible was treated with a single transcutaneous injection of freshly activated Fraction 2 PRGF. Clinical and radiographical follow-up was performed to assess healing.

### PRGF Preparation

For the preparation of PRGF, authorized commercial kit was used (KMU15, BTI Biotechnology Institute, Vitoria, Spain). Venous blood was extracted in tubes containing 3.8% sodium citrate as anticoagulant. The blood was centrifuged (System V, BTI Biotechnology Institute, Vitoria, Spain) for 8 min. The plasma column above the buffy coat was fractioned into Fraction 1 (F1) and Fraction 2 (F2). Fraction 2 is defined as the 2 mL of plasma located just above the buffy coat, and the rest of the plasma column is defined as fractioned as F1. To induce clot formation, 10% calcium chloride (PRGF activator) was used. The F2 was used to fill the bone defects for regeneration, and the F1 membrane was used to cover the surgical before flap closure. If needed in injectable form, the activated F2 was used immediately after adding the calcium chloride.

## 3. Results

Case description, surgical treatments, and follow-up data are presented for each case.

### 3.1. Case 1

A 79-year-old woman with a history of type II diabetes and a 57-pack-year smoking habit was seen on an emergency basis for swelling of the chin, unretractable pain, and numbness of the chin and lower lip. She denied any history of oral, IM, or IV bisphosphonates, likely due to her not having a family physician. Furthermore, she did not have a history of having taken steroids or other medications known to cause MRONJ. Implants had been placed between 2005 and 2014, with several previous implant failures having taken place. Clinical exam revealed submental induration, erythema, and edema. The intraoral exam revealed edema, erythema, exposed bone, purulence, and mobile implants 43 and 44. Panoramic radiograph had a mottled appearance with osteolysis to the inferior border and full thickness bone loss between implants 43 and 44 (Figure 1a).

Clavulin 500 mg/125 mg op tid was prescribed, and the Microbiology service was asked to consult. Two weeks later a crestal incision was made under local anesthesia (2% lidocaine with 1:200,000 epinephrine standard blocks and 3.6 cc and 4% articaine 1:200,000 epinepherine infiltration 1.8 cc) from 45–33 and implants 45,44,43,32 were removed with curettage and debridement of necrotic bone and sequestrae. A large parasymphysis sequestrae measuring 1.3 × 1.0 × 0.5 cm was removed and submitted for histopathologic analysis. At the time of surgery, full thickness involvement with perforation of the inferior border was noted at sites 44–43 (10 mm × 5 mm) and 32 (3 mm × 4 mm). Mandibular continuity was tenuous, yet intact. Necrotic gingival tissue was excised, and the entire defect was filled with PRGF Fraction 2, with Fraction 1 placed as a membrane prior to closure with 3-O polysorb. Where tension-free primary closure was not obtained (dehiscence measuring 16 mm × 8 mm approximately), the wound was left to close secondarily along the exposed PRGF Fraction 1 membrane. She was put on a liquid diet with nutrient supplements. Panoramic radiograph was taken immediately post-operatively (Figure 1b). The pathology report later revealed necrotic bone and was read as chronic suppurative osteomyelitis.

Two-week follow-up submental induration showed improvement. Intraoral exam revealed minimal edema and absence of inflammation. The PRGF Fraction 1 membrane was still exposed and intact, and the wound margins were observed to have contracted somewhat over the exposed membrane. At one-month follow-up, the patient presented to be happy and reported the absence of submental pain. Exam revealed resolution of swelling, and primary wound closure was noted, with an immature, healthy attached gingiva having regenerated over the crest of the anterior mandible. Her antibiotics were renewed. At 10-week post-operative follow-up, excellent, healthy, and thick attached gingiva was noted along the entire anterior reconstructed mandible. Panoramic radiograph showed excellent bone regeneration (Figure 1c). The patient was referred to her dentist to have a transitional denture fabricated, with continuation of soft diet. In coordination with Microbiology, oral antibiotics were ceased 1 month later (total of 4 months). The 26-week follow-up showed excellent bone regeneration, with no apparent loss of alveolar height (Figure 1d).

The patient presented for follow-up having broken the crowns on implant 34. Radiographic examination revealed fractured implant 34 (Figure 1e). Due to risk of pathological fracture, the fully integrated implant was cut using a round bur to the apical third under local anesthesia, and the resulting wound was treated with PRGF Fraction 2 and a Fraction 1 membrane over the wound. This healed uneventfully, and the patient underwent conservative implant reconstruction for implant overdenture at 52 weeks (Figure 1f). Of note is that the mandible was seemingly reconstructed to its entirety despite full thickness involvement with osteomyelitis, having undergone debridement and removal of large sequestrae and several failed implants. This was achieved with a single surgical intervention under local anesthesia despite the inability to achieve primary closure of the oral mucosa.

### 3.2. Case 2

An otherwise healthy 46-year-old male with no history of head and neck radiation and a one pack a day × 25-year smoking habit was referred for 3-week history of left lower lip numbness that had preceded recent pulpectomy of tooth 35. Teeth 36 and 37 had been previously endodontically treated. The endodontist did not wish to continue treatment of tooth 35 given that pulpectomy had not resolved the paresthesia.

Clinical exam revealed gross caries and temporary filling on tooth 37, leaving no restorative ferrule and an isolated 12+ mm pocket on the mesiobuccal aspect of tooth 36. Teeth 35, 36, and 37 were profoundly sensitive to percussion and periapical palpation. A periapical radiograph provided by the dentist (Figure 2a) demonstrated periapical radiolucencies of 37 roots and a diffuse radiolucency of 36 mesiobuccal root.

Panoramic radiograph (Figure 2b) showed a well-defined radiolucency extending from 36 medial and distal roots to the apex of 35 and involving the mental foramen. Cone beam computed tomography (CBCT) obtained by the endodontist confirmed the presence of sequestrum of the buccal cortex from 35–36 and surrounding the mental foramen (Figure 2c,d). Nerve testing revealed profound grade III paresthesia of the lower left inferior alveolar nerve. Diagnosis was osteomyelitis secondary to fractured 36, with involvement of teeth 35–36 and the inferior alveolar nerve causing sequestra formation, grade III paresthesia, non-vital 35, and unrestorable 37.

After discussion with the patient, he elected to undergo extraction of teeth 35, 36, and 37 with sequestrectomy/curettage using PRGF as an adjunct. He was prescribed an 8-week course of amoxicillin 500 mg tid (replacing clindamycin) for osteomyelitis and Lyrica 50 mg op tid × 2 months for the inferior alveolar nerve paresthesia. A consultation was also requested from the Microbiology Service.

Two weeks later teeth 35–37 were removed under local anesthesia (1.8 cc of 2% lidocaine with 1:200,000 epinephrine standard block and 1.8 cc of 4% articaine 1:200,000 epinephrine infiltration). Of note, clinically what had appeared on CBCT as a sequestrum was not mobile and still attached to the mandibular body. The lateral aspect of the mandible was hypodense, with a vermicular perforation of the lateral mandible body marking the outline of what had been considered radiographically to be a sequestrum. The mental nerve was Identified and protected, and the cavernous outline of hypodense bone was curetted of granulation tissue. Further examination confirmed that there was no sequestrum, but rather what appeared clinically to be a developing sequestrum. Freshly activated liquid PRGF F2 was then injected along the perforated cortical outline of the hypodense bone and around the inferior alveolar nerve and mental foramen using a tuberculin syringe. F2 coagulum was placed in the sockets of 35, 36, and 37 and along the lateral aspect of the mandible. Finally, F1 membranes were placed on lateral aspect of the mandible extending over the crest and below the lingual aspect of the flap. The gingival was re-approximated, with exposed F1 membrane over the sockets 35, 36, and 37, hence, without complete primary closure (Figure 2e). Two weeks later, the patient reported improvement in his paresthesia, and clinically, this was upgraded to grade II paresthesia. The surgical site was without pain, and there were no signs nor symptoms of infection. At his one month follow-up, his inferior alveolar nerve paresthesia had continued to improve and was assessed as grade I paresthesia clinically. Exam revealed excellent healing with mature attached gingiva overlying the extraction sockets, i.e., no dehiscence, no pain, and no signs nor symptoms of infection, no evidence of worsening nor any evidence of progression or sequestrum formation were observed. At 6 weeks, the patient reported that inferior alveolar nerve function was almost normal. In coordination with Microbiology, the patient was told to discontinue amoxicillin at the end of his 2-month course. He was administered a 4-week tapering regime for Lyrica. At 10-week follow-up, the paresthesia continued to resolve, and there was absence of clinical signs of osteomyelitis. At 16 weeks, 6 weeks after discontinuing antibiotics, there was again absence of any signs of osteomyelitis. At 20 weeks, paresthesia had completely resolved. At six months post-operative his radiograph showed further osseous healing (Figure 2f).

Implants were placed (8.5 mm length) at sites 35 and 36 at 52 weeks post-operative using BTI implant protocol including the treatment of the implant osteotomy with unactivated F2 immediately prior to implant placement (Figure 3a). At the time of implant placement, quality 1 bone was encountered, and insertion torque of the implants exceeded 35 NCm, allowing for a single-stage procedure. At 3.5-month follow-up, implants were integrated. There was an absence of lymphadenopathy, no signs nor symptoms of recurrence. Excellent attached gingiva was present around the implants without inflammation and continued absence of pain and tenderness of the alveolus and vestibule. Panoramic radiograph showed continued osseous consolidation and an absence of osteomyelitis recurrence. (Figure 3b). Radiographs in this case showed excellent bone regeneration of the left mandible without loss of height after conservative surgical management of osteomyelitis in a single intervention under local anesthesia.

### 3.3. Case 3

An otherwise healthy 58-year-old male with a one pack a day × 40 year smoking habit was referred for removal of remaining teeth and management of a large cyst (Figure 4a) associated with tooth 43. The patient expressed an interest in having dental implants placed for mandibular overdentures. Clinical exam revealed a hopeless dentition, aside from teeth 24 and 25 with a guarded prognosis. Pre-op CT showed an absence of buccal bone in the 45–45 region. Initial management included enucleation and excisional biopsy of the cyst associated with the apex of tooth 44, clearance of all teeth except 24, 25, 33, and 43, regeneration of the cyst, and preservation of sockets using PRGF. The procedure was performed under local anesthesia (5.4 cc 2% lidocaine 1:200,000 epinephrine standard and infraorbital blocks and infiltration with 3.6 cc 4% articaine 1:200,000 epinephrine). At time of enucleation, the buccal bone was noted to be a fraction of a millimeter in thickness on the buccal aspect of 43–45 region, as seen on pre-operative CT (Figure 4b) and as is common with odontogenic cysts. PRGF F2 was placed in the osseous defect and F1 membrane at the crest. An immediate post-operative radiograph was taken, showing the increased size of the defect from initial presentation (Figure 4c). Histopathological analysis confirmed the cyst was a radicular cyst. The six-month follow-up showed an impressive, complete regeneration of the alveolar bone, with virtually no loss of height and clinically complete regeneration of pre-operative alveolar width (Figure 4d). The remaining teeth 33 and 43 were removed with simultaneous placement of four mandibular implants for removable overdenture 8 months following enucleation of the cyst (Figure 4e). The implants were loaded 4.5 months later. Bone levels are excellent with no bone loss at 14-month follow-up (Figure 4f). 

### 3.4. Case 4

A healthy 57-year-old male was referred for jaw pain localized to the inferior border of the right mandibular parasymphysis region (Figure 5a). The pain had increased in intensity over the previous month. The patient denied trauma, injury, or cut to the epidermis in the region and denied any tooth pain. There were no nodes palpated on head and neck exam, with only a tenderness and slight hard swelling of the inferior border of mandible in the right parasymphysis region. Intraoral exam was normal, and panoramic radiograph was normal except for a well-defined irregular radiolucency of the right parasymphysis inferior cortex surrounding a small radiopaque bony island. There were no caries and no evident dental pathology. Cone beam CT revealed an 18 mm × 7 mm radiolucency surrounding an approximately 6 mm × 3 mm bony sequestrum (Figure 5b–e). Diagnosis of aseptic osteomyelitis was performed, despite the patient not being an adolescent. He was prescribed amoxicillin 500 mg tix for 2 months and provided a referral to the Microbiology consult service.

He was offered, as an adjunct to conventional management, injection of PRGF freshly activated F2 in a supraperiosteal plane via aseptic, extraoral approach, in the hopes that increased re-vascularization of the typically hypovascular inferior cortex could assist in resolution of his condition. Using aseptic technique, the parasymphysis region was infiltration with 1 cc of 2% lidocaine prior to injection of 4 cc of freshly activated F2 was injected via extraoral approach in the region of the right parasymphysis inferior border one week after antibiotics were initiated. Skin was prepped with chlorhexidine, and the inferior border sounded with a needle prior to each injection. Four injections of 0.75 cc each were administered at the periphery and a fifth at the center of the bony hard inferior border swelling. At his 2-week follow-up appointment, the patient reported that the pain had completely resolved within 5 days of the injections. Exam was normal at that time, with a possible reduction in bony hard swelling at the inferior border of his right mandible. A follow-up Cone Beam CT was performed at 5 months, and this showed osseous regeneration and loss of the radiopaque bony island within the radiolucency (Figure 5f and Figure 6a–d).

### 3.5. Case 5

An otherwise healthy 54-year-old male was referred for extraction of impacted tooth 38. He presented with trismus, left buccal swelling, and pain following two courses of antibiotics provided by his dentist. Exam revealed a fistula at the crest of edentulous posterior left mandibular segment. Panoramic exam revealed an impacted 38 with significant osseous destruction (Figure 7a). Cone Beam CT was performed and showed extensive osteolysis and thin residual inferior cortex (Figure 7b). He was put on liquid diet and pathologic mandible fracture precautions. Clavulin 500/125 mg op tid × 10 days was prescribed, and he was scheduled for extraction of tooth 38. Treatment with PRGF was recommended to speed osseous healing in the hopes of further minimizing the likelihood of pathologic fracture. One week later, he presented for removal of the tooth, complaining of new numbness of the lower lip. The tooth was removed uneventfully under local anesthesia (2% lidocaine 1:200,000 epi and 1.8 cc standard block and 4% articaine 1:200,000 epi infiltration), and the associated cyst-like lining was submitted for histopathological analysis, which later was interpreted by the pathologist to be a dentigerous cyst. PRGF F2 clot was used to entirely fill the osseous defect, and F1 membrane was placed over the mandibular crest (Figure 7c). He was prescribed Lyrica for the pre-operative inferior alveolar nerve paresthesia. At his 2-week follow-up, the patient reported that he did not take the Lyrica as he reported that the paresthesia had resolved within 24 h of the surgical procedure. Unfortunately, he did not present for his subsequent follow-up appointment and was lost to follow-up.

## 4. Discussion

This study has found that the adjunctive use of PRGF in the surgical treatment of osteomyelitis of the mandible has resulted in complete regeneration and healing prior to implant supported overdenture rehabilitation. Moreover, it has resulted in the complete regeneration of a large osseous defect after the surgical treatment of radicular cyst, allowing for the placement of implant-supported prosthesis.

The success criteria for successful management of osteomyelitis has traditionally been maintenance of function and continuity, resolution of pain, obtaining soft tissue coverage of exposed bone, and radiographic evidence of resolution and osseous repair, with absence of disease recurrence [30,31]. Similarly, the success criteria for management of medium and large odontogenic cysts has been maintenance of function and continuity, radiographic evidence of typically a combination of osseous repair/regeneration and absence of recurrence [32,33]. Conventional treatment of osteomyelitis would commonly involve sequestrectomy, decortication, and/or saucerization. This typically results in reduced alveolar volume in the months following surgery. Furthermore, conventional treatment of large odontogenic cysts typically involves curettage, and at times, peripheral ostectomy is used. Where bone is thin, even conservative curettage will often result in loss of alveolar height and width following surgery, which may make ideal implant placement problematic or impossible.

With the advent of tissue engineering and adjunctive methods to enhance tissue regeneration, these traditional definitions of success will be challenged to include successful regeneration of osseous volume, specifically bony width and height, to allow for functional rehabilitation using dental implants where desired and where appropriate. In this case report, the authors present a simple, non-invasive adjunct to management of commonly encountered pathology that appears also to assist in disease resolution in the case of osteomyelitis, with possibly a more conservative surgical approach, without the use of decortication or saucerization, and without the removal of a nascent sequestrum (Case 2) . Furthermore, the use of adjunctive PRGF appears to aid significantly in the regeneration of pre-surgical osseous volume in cases where conventional treatment would normally result in significant tissue loss, often precluding or resulting in clinical compromises in successful implant rehabilitation.

PRGF was introduced in 1999, and it has been characterized by the absence of leukocytes and the moderate concentration of platelets [34]. After activation with calcium ions [35], a dense fibrin rich matrix is formed. When placed in an osseous defect, cells will invade and migrate along the PRGF scaffold [21,36,37,38], to achieve healing [38,39]. PRGF has been demonstrated to support re-vascularization of mucosa and bone, modulate the inflammation, significantly speed bone healing, and maintain bone volume after surgical interventions [35,40,41,42,43]. For example, impaired angiogenesis by bisphosphonates plays a pivotal role in the inhibition of cell proliferation and the development of MRONJ [44,45]. Supporting this idea, reduced circulating growth factors have been measured in patients under bisphosphonate therapy [46,47]. PRGF has also the capacity through the provision of cell signaling molecules to promote angiogenesis, enhance collagen production, reduce inflammation, and induce cell differentiation [48,49,50,51]. An in vitro study has shown the cytoprotective effect of PRGF against the side effects of zoledronic acids through the inhibition of inflammation and apoptosis [51]. Furthermore, PRGF has been effective in the reduction of pain and tumefaction [35,52]. The clinical evidence has shown that the combination of surgical osteotomy (often resulting in large bone defect) and PRGF application to treat MRONJ lesions has been effective in achieving soft tissue closure and bone regeneration [7,53,54,55,56,57].

Another interesting aspect of PRGF is its antimicrobial properties that would be useful to reduce the risk of post-operative infection [58,59,60]. Drago et al. have shown that PRGF inhibited the growth of *Enterococcus faecalis*, *Candida albicans*, *Streptococcus agalactiae*, and *Streptococcus oralis* [60]. It appears that calcium activation of platelet-rich plasma is necessary to produce this antimicrobial effect [58]. In this case report, similarly, calcium-activated PRGF was used, and the clinical results likely benefited from the same antimicrobial effects, which may have been particularly important in the conservative treatment of osteomyelitis cases.

The adjunctive use of PRGF has enhanced the healing of injured inferior alveolar nerve in a patient with osteomyelitis. This is similar to other studies reporting positive effect of PRGF, in the treatment of MRONJ lesions, on the remission of symptoms and impaired function of the inferior alveolar nerve [53,55]. Plasma rich in growth factors has been successfully applied in different forms (injectable liquid, scaffold, and membrane) to treat neuropathies and injured peripheral nerves (traumatic or non-traumatic) [61,62,63]. This efficacy of PRGF could be related to the acceleration of recovery through the modulation of inflammation, activation of the neo-angiogenesis, activation of the Schwann cell, and macrophage polarization [62,64,65,66,67,68]. Thus, PRGF could contribute to functional nerve recovery by the resolution of inflammation and fibrogenesis. Moreover, a growing body of research supports the therapeutic use of PRGF to enhance the robust intrinsic nerve repair process and overcome an inhibitory post-traumatic and neuropathic microenvironment through the delivery of neurotrophic and neurotrophic factors [69]. PRGF could offer two vital benefits: first, to produce a prolonged and gradual stimulatory effect (antiapoptotic, chemotactic, and anti-inflammatory supportive signals), and second to function as a transient scaffold that serves as a guide to new axonal sprouts [62,64,69]. By studying the early healing of injured peripheral nerve, Torul et al. have observed that PRGF enhanced nerve healing while leukocyte-rich, platelet-rich fibrin has a limited effect [70].

The results of this case report are consistent with the observation of other studies treating non-oncological osseous pathology [7,53,54,55,56,57], where the adjunctive use of PRGF has enhanced the healing of large osseous defects and the injured inferior alveolar nerve. PRGF should be considered in the management of non-neoplastic pathology where increased vascularization is desirable or when conventional treatment would normally be expected to result in osseous tissue loss, to limit morbidity and enhance functional rehabilitation, with dental implants where appropriate.

## 5. Conclusions

Conventional treatment of osteomyelitis is challenging and often results in unacceptable morbidity and functional compromise. There is a clear need to improve predictability of treatment, particularly if the morbidity and functional outcomes can be improved. Similarly, odontogenic cysts, while predictably treated, can often result in loss of function or osseous defects. Furthermore, there is a role for adjuncts in improving functional rehabilitation in the treatment of odontogenic cysts that cause localized osseous tissue destruction.

While further controlled studies are necessary, PRGF is a simple and safe adjunct to conventional surgical treatment of osteomyelitis and large odontogenic cysts that may be easily performed at the time of surgery to enhance host response and to maintain bone volume.

## Figures and Tables

**Figure 1 dentistry-11-00184-f001:**
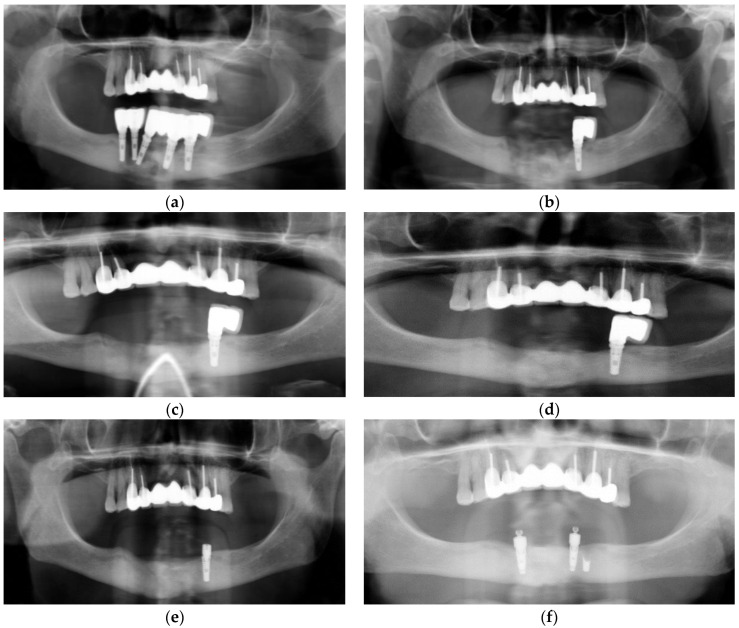
Case 1: (**a**) Extensive osteolysis of implants 45,44,43, and 32 extending to the inferior border of the mandible; (**b**) Immediate post-operative panorex following curettage and debridement with removal of sequestrae. Note that extensive corticotomy and saucerization were not performed; (**c**) The 10-week post-operative panoramic radiograph. Note that bone regeneration with no apparent loss of osseous height compared to pre-operative panoramic radiograph was observed; (**d**) The 26-week follow-up panoramic radiograph showing bone consolidation; (**e**) Fractured implant 34; (**f**) At 52 weeks after debridement—Implant placement for conservative over denture rehabilitation.

**Figure 2 dentistry-11-00184-f002:**
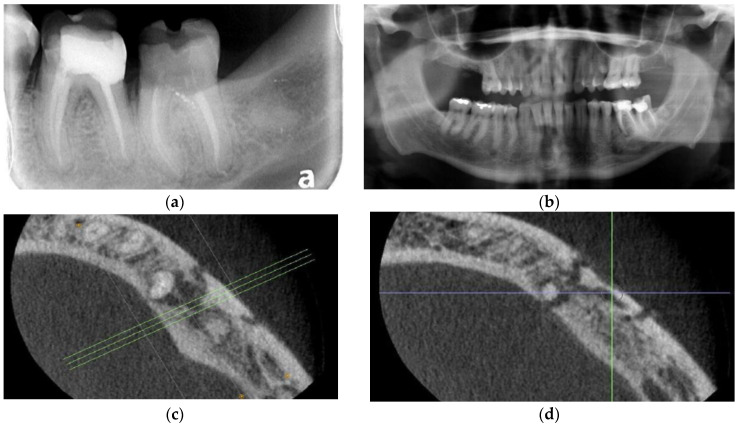
Case 2: (**a**) Periapical radiograph of teeth 36 and 37. Note caries at 35D and ill-defined radiolucency associated with recently endodontically treated 36 and 37; (**b**) Panoramic radiograph of teeth 33–37 showing osteolysis and ill-defined radiolucency associated with teeth 35–36; (**c**) Axial view of CBCT right mandible showing osteolysis and apparent sequestrum forming on the buccal cortex associated with teeth 35 and 36 and involving the mental foramen; (**d**) Axial view of CBCT right mandible showing osteolysis and apparent sequestrum forming on the buccal cortex associated with teeth 35 and 36 and involving the mental foramen; (**e**) Immediate post-operative panoramic radiograph showing sequestrum formation overlying the mental foramen; (**f**) The 6-month post-operative panoramic radiograph. Note increased osseous radiodensity and evidence of resolution, rapid osseous regeneration and maintenance of pre-operative alveolar height.

**Figure 3 dentistry-11-00184-f003:**
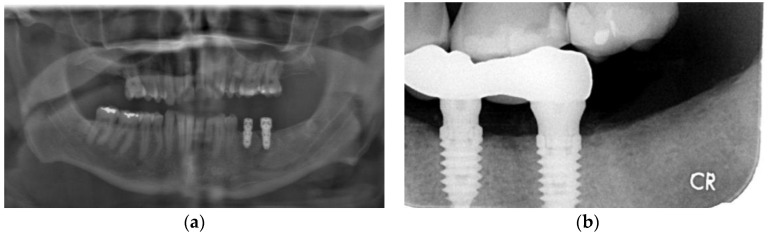
Case 2. Treatment with dental implants: (**a**) Implants placed 52 weeks after conservative PRGF treatment of osteolysis mandible; (**b**) Restored implants 35 and 36 after 4 months of implant placement.

**Figure 4 dentistry-11-00184-f004:**
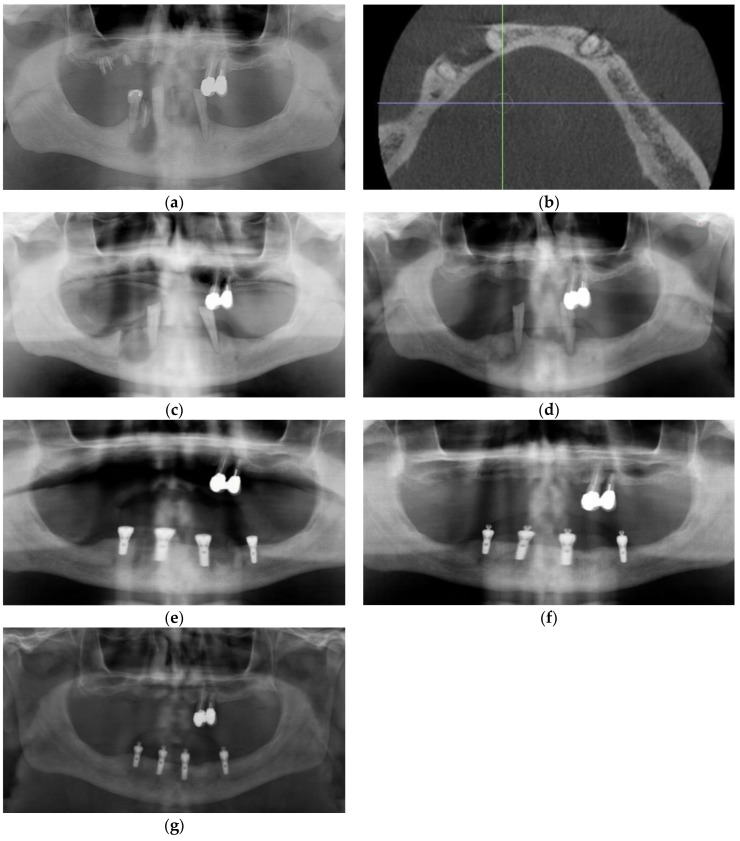
Case 3: (**a**) Pre-operative panoramic radiograph showing a large odontogenic cyst at the apex of tooth 44; (**b**) CBCT showing extensive thinning and perforation of the buccal plate; (**c**) Immediate post-operative panoramic radiograph; (**d**) Panoramic radiographs showing alveolar regeneration 6 months following radicular cyst enucleation; (**e**) Panoramic radiograph immediately following implant placement (8 months following cyst enucleation); (**f**) Panoramic radiograph taken 6 months following implant placement. Note that height of bone and placement of implants are in the ideal location, with implant 44 placed above the inferior the mental foramen; (**g**) Panoramic radiograph taken 14 months following implant placement.

**Figure 5 dentistry-11-00184-f005:**
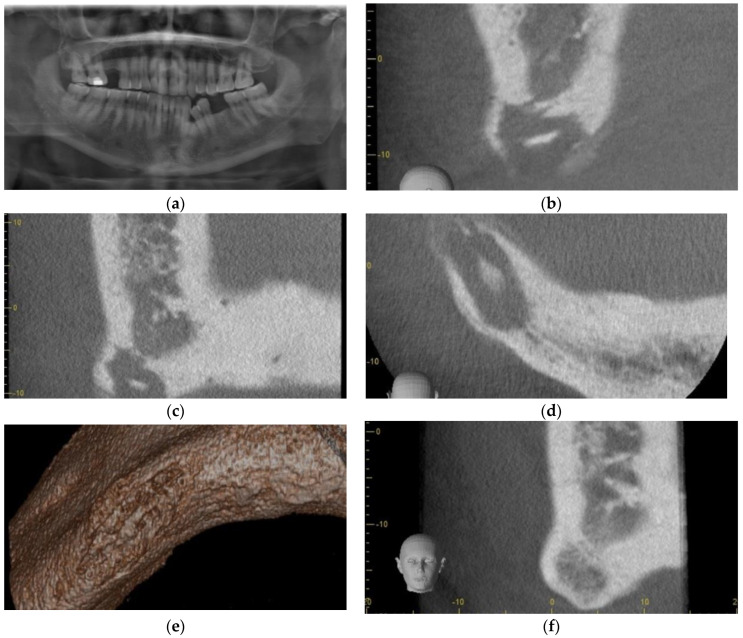
Case 4: (**a**) Panoramic radiograph, note inferior cortex right parasymphysis osteolysis; (**b**) Sagittal section Cone Beam CT scan showing osteolysis of dense cortical bone, extending to the inferior border, with central sequestrum; (**c**) Coronal section Cone Beam CT showing osteolysis of dense cortical bone, extending to the inferior border, with central sequestrum; (**d**) Axial section Cone Beam CT showing osteolysis of dense cortical bone, extending to the inferior border, with central sequestrum; (**e**) 3D reconstruction of Cone Beam CT showing osteolysis of the inferior border of the mandible; (**f**) Coronal section CBCT 5 months post-PRGF treatment, and osseous regeneration of the Inferior cortical bone Is readily apparent.

**Figure 6 dentistry-11-00184-f006:**
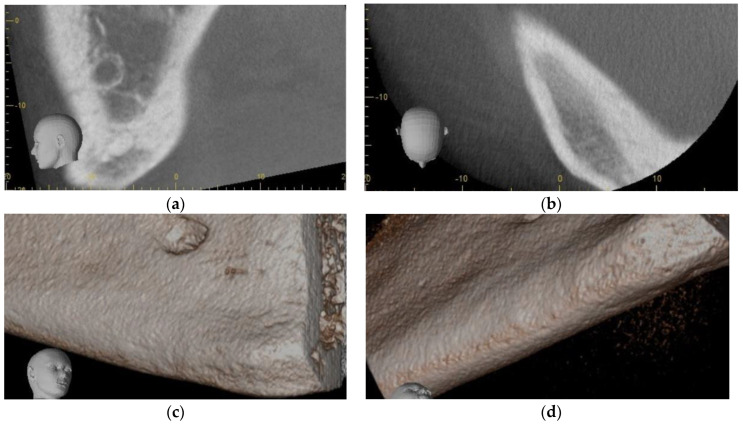
Case 4 follow-up: (**a**) Sagittal section of CBCT 5 months post-PRGF treatment, and osseous regeneration of the inferior cortical bone is readily apparent; (**b**) Axial section CBCT 5 months post-PRGF treatment, and osseous regeneration of the inferior cortical bone is readily apparent; (**c**) 3D reconstruction of CBCT 5 months post-PRGF treatment, and osseous regeneration of the inferior cortical bone is readily apparent; (**d**) 3D reconstruction of CBCT 5 months post-PRGF treatment, and osseous regeneration of the inferior cortical bone is readily apparent.

**Figure 7 dentistry-11-00184-f007:**
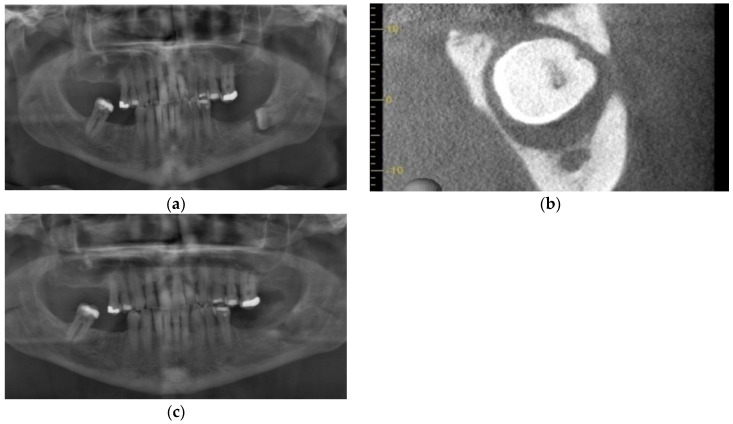
Case 5: (**a**) Impacted tooth 38 with extensive peri-coronal bone loss approaching the inferior alveolar nerve; (**b**) Pre-operative CBCT showing buccal plate and alveolar crest perforation and extreme thinning of the lingual cortex, presenting risk of pathologic fracture; (**c**) Immediate post-operative panoramic radiograph.

## Data Availability

All available data was reported in this manuscript.

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
