# Peer review of "Adjunctive Plasma Rich in Growth Factors in the Treatment of Osteomyelitis and Large Odontogenic Cysts Prior to Successful Implant Rehabilitation: Case Report"

_dentistry, 2023, doi:10.3390/dj11080184_

Round 1
Reviewer 1 Report
Dear Authors,
I have read your paper entitled: "Adjunctive PRGF in the treatment of Osteomyelitis and large odontogenic cysts prior to successful implant rehabilitation: Case report".
1. Please use the Journal template!
2. Please modify your title, this is not a case report, while case series.
3. Please expand the affiliation of the authors following the Journal template.
4. Please add Keywords following the Journal template.
5. Please format the numeber of citations according to the Journal template.
6. Please add the etiopathological path of osteomyelitis in your Introduction.
7. When you write about MRONJ you have to briefly report all major causes of this adverse event such as oral surgery, endodontic treatment, non surgical periodontal therapy. In order to have solid References please add the following papers to your citations:
https://pubmed.ncbi.nlm.nih.gov/35300956/
https://pubmed.ncbi.nlm.nih.gov/36403223/
https://pubmed.ncbi.nlm.nih.gov/29479739/
https://pubmed.ncbi.nlm.nih.gov/37048016/
8. Please end your Introduction by writing clearly the purpose of your study.
9. In Matherials and Methods, please add the inclusion criteria, the exclusion criteria, the Local Ethical Committee Code and if the study was run according to the Helsinki Declaration. Explain here your protocol to obtain PRGF (paragraph 2.6 fits better in M&M section).
10. In case 1, please clarify the pharmacological anamnesis of the patient, in particular the supposed absence of drugs related to MRONJ (not only bisphosphonates). Please clarify also if the patient underwent or not radiotherapy of the head&neck district.
11. Do the authors have any clinical pictures of the surgery and of the follow up period?
12. In case 2, please clarify the pharmacological hystory and also if the patient underwent or not radiotherapy of the head&neck district.
13. Please expand your Conclusion.
Author Response
We thank the reviewer for his/her comments that added clarity to the manuscript.
- Please use the Journal template!
Done
- Please modify your title, this is not a case report, while case series.
Done
- Please expand the affiliation of the authors following the Journal template.
Done
- Please add Keywords following the Journal template.
Done
- Please format the numeber of citations according to the Journal template.
Done
- Please add the etiopathological path of osteomyelitis in your Introduction.
Done
- When you write about MRONJ you have to briefly report all major causes of this adverse event such as oral surgery, endodontic treatment, non surgical periodontal therapy. In order to have solid References please add the following papers to your citations:
https://pubmed.ncbi.nlm.nih.gov/35300956/
https://pubmed.ncbi.nlm.nih.gov/36403223/
https://pubmed.ncbi.nlm.nih.gov/29479739/
https://pubmed.ncbi.nlm.nih.gov/37048016/
The following has been added to the text: Unfortunately, bisphosphonates are well known to predispose patients to medication related osteonecrosis of the jaw (MRONJ), particularly in the context of surgical, endodontic or periodontal manipulation.[6-9]
The following references have been added:
- D'Agostino, S.; Valentini, G.; Dolci, M.; Ferrara, E. Potential Relationship between Poor Oral Hygiene and MRONJ: An Observational Retrospective Study. Int J Environ Res Public Health 2023, 20, doi:10.3390/ijerph20075402.
- Karna, H.; Gonzalez, J.; Radia, H.S.; Sedghizadeh, P.P.; Enciso, R. Risk-reductive dental strategies for medication related osteonecrosis of the jaw among cancer patients: A systematic review with meta-analyses. Oral Oncol 2018, 85, 15-23, doi:10.1016/j.oraloncology.2018.08.003.
- Ruggiero, S.L.; Dodson, T.B.; Aghaloo, T.; Carlson, E.R.; Ward, B.B.; Kademani, D. American Association of Oral and Maxillofacial Surgeons' Position Paper on Medication-Related Osteonecrosis of the Jaws-2022 Update. J Oral Maxillofac Surg 2022, 80, 920-943, doi:10.1016/j.joms.2022.02.008.
- Tempesta, A.; Capodiferro, S.; Di Nanna, S.; D'Agostino, S.; Dolci, M.; Scarano, A.; Gambarini, G.; Maiorano, E.; Favia, G.; Limongelli, L. Medication-related osteonecrosis of the jaw triggered by endodontic failure in oncologic patients. Oral Dis 2022, doi:10.1111/odi.14449.
- Please end your Introduction by writing clearly the purpose of your study.
Done
- In Matherials and Methods, please add the inclusion criteria, the exclusion criteria, the Local Ethical Committee Code and if the study was run according to the Helsinki Declaration. Explain here your protocol to obtain PRGF (paragraph 2.6 fits better in M&M section).
Done
- In case 1, please clarify the pharmacological anamnesis of the patient, in particular the supposed absence of drugs related to MRONJ (not only bisphosphonates). Please clarify also if the patient underwent or not radiotherapy of the head&neck district.
Done
- Do the authors have any clinical pictures of the surgery and of the follow up period?
This is a case series of a retrospective nature and the dependence on the available data in patients’ record is clear. We have added follow-up images for all cases. Case 5 was lost to follow-up and we have only the immediate post-operative radiograph.
- In case 2, please clarify the pharmacological history and also if the patient underwent or not radiotherapy of the head&neck district.
Done
- Please expand your Conclusion.
Done
We thank the reviewer for his/her opinion and help.
Marc DuVal
Reviewer 2 Report
Dear colleagues!
Your message is interesting and relevant for modern dentistry.
As a criticism, I want to draw attention to the low reliability of the 2D image. Since you are showing CT and 3D reconstruction, the illustrative material should be uniform.
I also draw attention to the need to specify the discussion area on all images.
Case 1 - what dosage of local anesthetic and vasoconstrictor was used?
In general, the design of clinical cases is correct.
Author Response
Dear Colleague,
Thank you for your review of our submission.
Dear colleagues!
Your message is interesting and relevant for modern dentistry.
We thank the reviewer for his/her comments that added clarity to the manuscript.
As a criticism, I want to draw attention to the low reliability of the 2D image. Since you are showing CT and 3D reconstruction, the illustrative material should be uniform.
This is a case series of a retrospective nature and the dependence on the available data in patients’ record is clear. We agree with the reviewer that 3D imaging is more informative and for that we have included it when it has been available. To assess healing and changes in bone defects, 2D radiographs have still in use in the clinical practice. At the time of treatment, I did not know we would be submitting a series of cases, having not anticipated the results. This aspect will be taken in consideration in the next studies on this topic.
I also draw attention to the need to specify the discussion area on all images.
We thank the reviewer for this comment. We have provided more detiales in the figure footnotes when needed. The Journal has it’s own format where one single section for discussion is indicated.
Case 1 - what dosage of local anesthetic and vasoconstrictor was used?
This information has been added to Case 1: (2% lidocaine with 1:200,000 epinephrine standard blocks and 3.6cc and 4% articaine 1:200,000 epinepherine infiltration1.8cc). Similar precision was added to the remaining cases 2-5.
In general, the design of clinical cases is correct.
We thank the reviewer for his/her opinion and help.
Thank you,
Marc DuVal